# Magneto-Plasmonic Nanoparticles Generated by Laser Ablation of Layered Fe/Au and Fe/Au/Fe Composite Films for SERS Application

**Lina Mikoliunaite** [1,2,*], **Evaldas Stankevičius** [3], **Sonata Adomavičiūtė-Grabusovė** [4], **Vita Petrikaitė** [3], **Romualdas Trusovas** [3], **Martynas Talaikis** [1], **Martynas Skapas** [5], **Agnė Zdaniauskienė** [1], **Algirdas Selskis** [5], **Valdas Šablinskas** [4] and **Gediminas Niaura** [1,*]

1   Department of Organic Chemistry, Center for Physical Sciences and Technology (FTMC), Sauletekio Av. 3, LT-10257 Vilnius, Lithuania; martynas.talaikis@ftmc.lt (M.T.); agne.zdaniauskiene@ftmc.lt (A.Z.)
2   Department of Physical Chemistry, Faculty of Chemistry and Geosciences, Vilnius University, Naugarduko Str. 24, LT-03225 Vilnius, Lithuania
3   Department of Laser Technologies, Center for Physical Sciences and Technology (FTMC), Savanoriu Av. 231, LT-02300 Vilnius, Lithuania; estankevicius@ftmc.lt (E.S.); vita.petrikaite@ftmc.lt (V.P.); romualdas.trusovas@ftmc.lt (R.T.)
4   Institute of Chemical Physics, Faculty of Physics, Vilnius University, Sauletekio Av. 3, LT-10257 Vilnius, Lithuania; sonata.adomaviciute@ff.vu.lt (S.A.-G.); valdas.sablinskas@ff.vu.lt (V.Š.)
5   Department of Characterization of Materials, Center for Physical Sciences and Technology (FTMC), Sauletekio Av. 3, LT-10257 Vilnius, Lithuania; martynas.skapas@ftmc.lt (M.S.); algirdas.selskis@ftmc.lt (A.S.)
*   Correspondence: lina.mikoliunaite@ftmc.lt (L.M.); gediminas.niaura@ftmc.lt (G.N.)

**Abstract:** Magneto-plasmonic nanoparticles were fabricated using a 1064 nm picosecond-pulsed laser for ablation of Fe/Au and Fe/Au/Fe composite thin films in acetone. Nanoparticles were characterized by electron microscopy, ultraviolet-visible (UV-VIS) absorption, and Raman spectroscopy. Hybrid nanoparticles were arranged on an aluminum substrate by a magnetic field for application in surface-enhanced Raman spectroscopy (SERS). Transmission electron microscopy and energy dispersive spectroscopy analysis revealed the spherical core-shell (Au-Fe) structure of nanoparticles. Raman spectroscopy of bare magneto-plasmonic nanoparticles confirmed the presence of magnetite ($Fe_3O_4$) without any impurities from maghemite or hematite. In addition, resonantly enhanced carbon-based bands were detected in Raman spectra. Plasmonic properties of hybrid nanoparticles were probed by SERS using the adsorbed biomolecule adenine. Based on analysis of experimental spectra and density functional theory modeling, the difference in SERS spectra of adsorbed adenine on laser-ablated Au and magneto-plasmonic nanoparticles was explained by the binding of adenine to the $Fe_3O_4$ structure at hybrid nanoparticles. The hybrid nanoparticles are free from organic stabilizers, and because of the biocompatibility of the magnetic shell and SERS activity of the plasmonic gold core, they can be widely applied in the construction of biosensors and biomedicine applications.

**Keywords:** magneto-plasmonic nanoparticles; laser ablation; SERS; thin film; magnetite; gold; core-shell nanoparticles

## 1. Introduction

Magneto-plasmonic nanoparticles are composites that combine magnetic and plasmonic materials in a confined nanoscale area and simultaneously exhibit magnetic and plasmonic properties [1–4]. They typically use Fe, Co, or Ni-based magnetic materials. The most popular among them are iron and magnetite. $Fe_3O_4$ is known to be nontoxic, biocompatible, and possesses an inducible magnetic moment; thus, it can be used for hyperthermia, targeted drug delivery, extraction of biomolecules, lab-on-a-chip construction, and other biomedical applications [5–8]. A noble metal (Au, Ag, or Pt) that separately could be used for resonance energy processes such as nanometal surface energy transfer

(NSET), fluorescence resonance energy transfer (FRET), cascade energy transfer (CET), metal-enhanced fluorescence (MEF), plasmon-induced resonance energy transfer (PIRET), and surface-enhanced Raman scattering (SERS) [9,10], adds to the system plasmonic components that could expand the application field of these nanostructures to stable molecule detection using SERS [11]. Such nanoparticles may consist of a noble metal layer/magnetic core or form an inverse structure [1]. Magneto-plasmonic nanoparticles are one of the new multifunctional materials for medical applications [3,4]. Nanocomposites of various combinations are used for phototherapy [12], as contrast agents in magnetic resonance imaging [13], brain disease treatment [14], cancer therapy and diagnostics [15], drug delivery, and hyperthermia applications [16]. For applications in diagnostics and even therapeutic fields, the plasmonic properties of these nanoparticles are of the utmost importance. The plasmonic nature of these nanoparticles leads to high molecular sensitivity in cases of application in SERS or high photothermal efficiency, while magnetic properties ensure control of the spatial position of nanoparticles.

Laser irradiation has proven to be a versatile tool for nanoparticle synthesis [17]. Laser ablation is a method for producing nanoparticles, nanowires, quantum dots, and core-shell nanoparticles [18–21]. Laser can be used to form nanoparticles by melting thin metal coatings or targets [19,20,22], ablation of metal granules [23], forming noble metal surfaces [24], or Si substrates that could later be covered with SERS active metal [25,26]. During the ablation process in gas, nanoparticles are created by the nucleation and growth of laser-evaporated species in a background gas. The ultra-fast vapor quenching is helpful for the production of high-purity nanoparticles in the quantum size range (<10 nm) [18]. One of the most commonly used methods for generating laser nanoparticles is pulsed laser ablation in liquids [27,28]. In this method, a pulsed laser beam is focused onto a solid target placed in a liquid medium that could be acetone, water, methyl methacrylate, or others [29]. Pulsed laser ablation in liquids is an attractive technique as it is chemically clean and requires no additional chemicals that, in some cases, may even be toxic [30,31]. This method is also attractive as the available target and fluid materials are extensive, including various metals and their alloys, semiconductors, oxides, alloys, and carbon allotropes.

Magneto-plasmonic Fe/Au alloy nanoparticles can be successfully ablated using solid Au-Fe targets in ethanol [32] and acetone [33]. Laser-induced generation of alloy and core-shell nanoparticles by ablation of multilayer films was previously investigated by Amendola et al. [34,35]. Alternatively, alloys of plasmonic/magnetic or plasmonic/plasmonic metals could also be used [25,29,36]. Nanosecond laser ablation yields higher relative concentrations of core-shell nanoparticles than picosecond ablation due to differences in laser radiation-plasma plume interaction time frames [35]. In the reference [34], nanoparticles were formed using a nanosecond laser of 1064 nm wavelength. Experiments were carried out with water and ethanol. Alloy nanoparticles were synthesized by ablation of Fe/Au layers of different thicknesses and compositions. Using ethanol as an ablation medium, alloy nanoparticles were formed with sizes of 7–8 nm with a standard deviation of 4–6 nm, whereas core-shell nanoparticles with a Fe core were achieved using water. Their results proved successful in the synthesis of alloy nanoparticles from films of 100–200 nm order thickness. Recently, authors demonstrated the formation of Fe/Au nanoparticles using laser ablation of bulk alloy targets [32]. Ablated nanoparticles or surface structures could later be decorated with chemically synthesized structures [24,37,38] or could be used as reductors of metal salts [22,39]. The produced nanoparticles were applied for rat blood analysis [23], detection of explosives [22,24,40], pyrromethene [41], antibiotics [26], pesticides [24,36,39], and many other molecules [38].

In our work, picosecond-laser ablation experiments for nanoparticle generation were carried out in acetone. It acts as a medium for the growth and stabilization of the nanoparticles. Acetone has been widely used in nanomaterial synthesis due to its ability to solubilize various metal precursors and provide a controlled environment for nanoparticle formation. The targets for ablation were created as composite coatings of varying thicknesses and layers of Fe/Au and Fe/Au/Fe. Various parameters were altered and optimized for the

synthesis of stable, plasmonic nanoparticles that also possess magnetic properties. These nanoparticles were applied for SERS analysis of the adenine molecule. The obtained results were compared to DFT-calculated spectra and showed that biological molecules preferably adsorb on magnetite surfaces.

## 2. Materials and Methods

### 2.1. Materials

Iron and gold targets for laser ablation were obtained from Micro-to-Nano (Haarlem, The Netherlands). Acetone ($\geq$99.9%), used for the ablation procedure, was purchased from Honeywell (Charlotte, NC, USA). Adenine ($\geq$99%), $D_2O$ (99.9 atom% D), and ethanol (99.5%) were purchased from Sigma-Aldrich (St. Louis, MO, USA), and deionized water (18.2 M$\Omega$·cm) was obtained from the Direct-Q 3UV purification system (St. Louis, MO, USA). 4-mercaptobenzoic acid (4-MBA) for SERS was from Thermo Scientific (Loughborough, UK).

### 2.2. Target Preparation

Gold and iron coatings (Fe/Au and Fe/Au/Fe) were prepared on a soda-lime glass substrate with a thickness of 1 mm using magnetron sputtering machine Q150T ES (Quorum, Laughton, UK) at room temperature in an Ar atmosphere of $10^{-3}$ bar with a deposition rate of 0.27 and 0.22 nm/s for gold and iron films, respectively. Gold and iron targets with a purity of 99.99% (Au) and 99.95% (Fe) were used. The thickness of thin metal films was controlled by varying the sputtering time. During deposition, the sample holder was rotating at 8 rpm. Before deposition, the soda-lime glass substrates were cleaned by washing with deionized water and holding for 2 h in concentrated $H_2SO_4$. After that, substrates were washed with water and sonicated two times in ethanol for 20 min. Finally, substrates were dried under nitrogen flow. The coating types and layer thicknesses are presented in Table 1.

**Table 1.** Composition of tested Fe/Au and Fe/Au/Fe layered coatings used in laser ablation experiments.

| Denotation | 1st Layer Fe Layer Thickness (nm) | 2nd Layer Au Layer Thickness (nm) | 3rd Layer Fe Layer Thickness (nm) |
|---|---|---|---|
| Fe50/Au150 | 50 | 150 | - |
| Fe100/Au150 | 100 | 150 | - |
| Fe50/Au150/Fe25 | 50 | 150 | 25 |
| Fe50/Au150/Fe50 | 50 | 150 | 50 |

### 2.3. Laser Ablation Procedure

A picosecond laser Atlantic (Ekspla, Vilnius, Lithuania) was used to generate magneto-plasmonic nanoparticles. A Galvoscanner hurrySCAN (ScanLab, Puchheim, Germany) with a 160 mm focal distance focusing objective was used for beam control. The pulse duration was 10 ps, the laser irradiation wavelength was 1064 nm, and the pulse repetition rate was 100 kHz. The laser beam was focused on a sample target inside the chamber, filled with acetone. The whole coated sample area was scanned with a laser beam using a hatch pattern. The distance between adjacent lines was 50 μm, scanning speed was 500 mm/s. The average laser irradiation power was 5 W, and the laser fluence was ~1.3 J/cm$^2$. The chosen fluence was similar to other authors' works: 1 J/cm$^2$ [42], 0.8 J/cm$^2$ [43], and 2.5 J/cm$^2$ [44]. The setup for laser ablation experiments is shown in Figure 1.

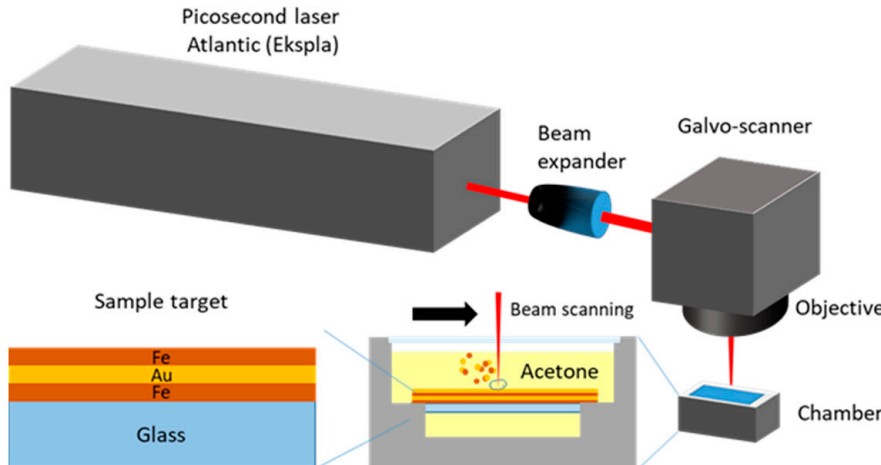

**Figure 1.** Experimental setup for the generation of magneto-plasmonic nanoparticles by laser ablation procedure.

Pure gold nanoparticles for comparison were also obtained using laser ablation. Gold targets were placed in a chamber filled with deionized water with 0.024 mM KCl. The final volume of the liquid was 19 mL. Ablation was conducted for 5 min, using 5 W of laser power.

### 2.4. Sample Characterization and Preparation for SERS Measurements

Ablated nanoparticles were characterized using UV-VIS-NIR, TEM, and SEM equipment. For extinction spectra measurements, a UV-VIS-NIR Lambda 1050 spectrometer (Perkin Elmer, Waltham, MA, USA) was employed (in the range 300–800 nm). Nanoparticles were imaged using scanning electron microscopy (SEM) using a dual-beam system, Helios Nanolab 650 (Thermo Scientific, Eindhoven, The Netherlands), and a transmission electron microscope, FEI Tecnai G2 F20 X-TWIN (Thermo Scientific, Eindhoven, The Netherlands).

The preparation of SERS substrates was performed as follows: The Nd magnet was wrapped in aluminum foil, cleaned with ethanol, dried, and immersed in a colloid of generated Fe/Au or Fe/Au/Fe nanoparticles for 60 s. The extracted magnet was washed with deionized water and dried. For comparison, pure gold nanoparticles were prepared by simply dropping a few drops of generated solution on aluminum foil and drying. The plasmonic properties of the generated nanoparticles were verified by measuring the SERS spectra of the test molecules (adenine and 4-mercaptobenzoic acid). The enhancement factor (*EF*) was calculated using 4-mercaptobenzoic acid. A 0.1 mM adenine solution in water was selected to evaluate the SERS performance for the analysis of biomolecules. One drop (25 µL) of a 0.1 mM adenine solution was dripped on the previously prepared SERS substrate. Spectra were measured by focusing a laser beam on the substrate with magneto-plasmonic nanoparticles in the presence of a water solution containing adenine.

The SERS spectra were measured using a MonoVista CRS+ spectrometer (S&I, Warstein, Germany) with an integrated optical microscope with a $100\times/0.80$ NA objective. An excitation wavelength of 632.8 nm was used, and the laser beam was focused on an area of approximately 1 $\mu m^2$ on the sample. The Raman spectra of the magneto-plasmonic nanoparticle substrate were recorded at a power of 0.8 mW. SERS spectra of the adsorbed adenine were acquired with a power of 2.5 mW. Raman spectra of adenine in solid state and 7.0 mM $H_2O$ and $D_2O$ solutions were recorded with the Raman spectrometer Hyper-Flux PRO Plus (Tornado Spectral Systems, Mississauga, ON, Canada), equipped with a thermoelectrically cooled fiber-optic cable and a 785 nm wavelength laser source. The powder sample was probed using 30 mW power and 10 s accumulation, while the adenine solutions were probed with 495 mW and 600 s. SERS enhancement factors at excitation

wavelengths 632.8, 785, and 830 nm were evaluated by using an inVia Raman spectrometer (Renishaw, Wotton-under-Edge, Gloucestershire, UK).

### 2.5. Density Functional Theory Modelling

Optimized geometry and numerical frequencies of adenine and adenine complexes in a vacuum were calculated using Orca 5.0.1 software [45] at the B3LYP theory level using the def2-TZVPP basis set. No imaginary frequencies were obtained.

## 3. Results

Laser ablation procedures for layered structures were conducted in deionized water, acetone, and isopropanol. In deionized water, the obtained nanoparticles aggregated in a few minutes after laser ablation and were not suitable for further investigation. Nanoparticle stability in isopropanol was moderate; however, the best results were obtained using acetone, so this solvent was chosen.

### 3.1. Structural and Magnetic Characterization of Magneto-Plasmonic Nanoparticles

The magneto-plasmonic nanoparticles ablated in acetone possessed magnetic properties, which can be confirmed by SEM images (Figure 2) of nanoparticles on aluminum foil substrates, oriented according to the magnetic field lines. From the images, different-sized nanoparticles are seen up to a few hundred nanometers.

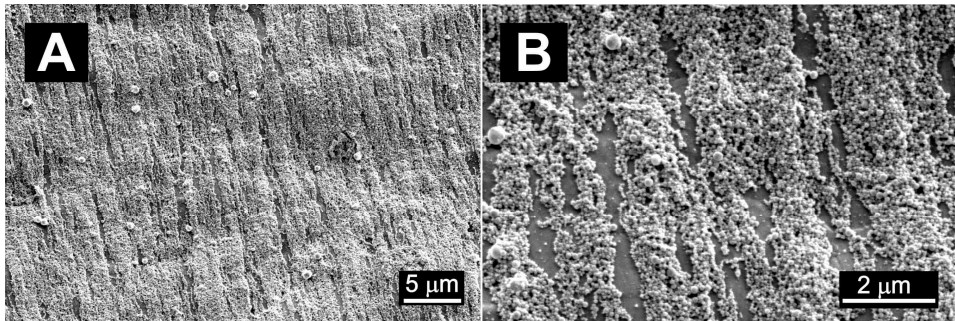

**Figure 2.** SEM image of magneto-plasmonic nanoparticles (Fe50/Au150/Fe25) concentrated by a magnetic field on an aluminum foil substrate. Two magnifications are presented: a scale bar of 5 μm (**A**) and 2 μm (**B**).

Figure 3 displays TEM measurements of the magneto-plasmonic nanoparticles generated from the Fe50/Au150/Fe25 and Fe50/Au150 targets. TEM imaging revealed the formation of spherical core-shells and homogeneous nanoparticles with a wide size distribution. The ImageJ program was used to calculate the size distribution of nanoparticles. The nanoparticles' diameter in samples Fe50/Au150/Fe25 was 46 ± 12 nm, while samples Fe50/Au150 showed 59 ± 28 nm. The energy dispersive spectroscopy (EDS) analysis (Figure 4B) confirmed that the core is composed of Au and is covered with a shell containing Fe and O. The high overall amount of oxygen in the sample is due to its abundance in the atmosphere and on the sample; however, a slightly higher percentage of O on the nanoparticles, especially at the sides of them, indicates the formation of magnetite. Thus, from the TEM images, the darker nanoparticles are composed of gold, while the lighter ones are iron oxide. The separately distributed homogeneous gold and iron nanoparticles are also registered. The amount of carbon in the sample is high due to the sample preparation for TEM; a copper mesh with a carbon layer was used for visualization, so detection of carbon in the ablated sample is hardly possible.

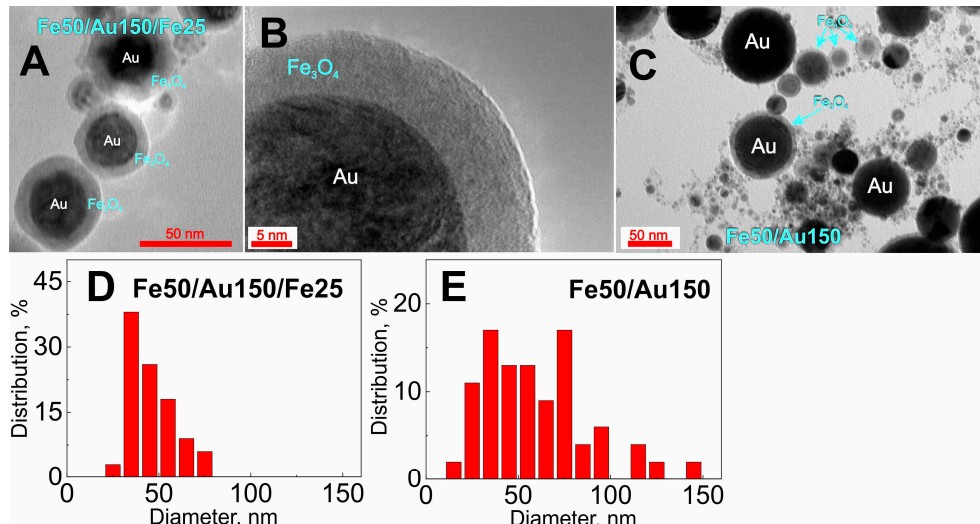

**Figure 3.** TEM images of magneto-plasmonic nanoparticles generated from Fe50/Au150/Fe25 targets at different magnifications: 50 nm scale bar (**A**) and 5 nm scale bar (**B**) and nanoparticles from Fe50/Au150 target (**C**). The size distribution of nanoparticles is presented below: for Fe50/Au150/Fe25 (**D**) and for Fe50/Au150 (**E**).

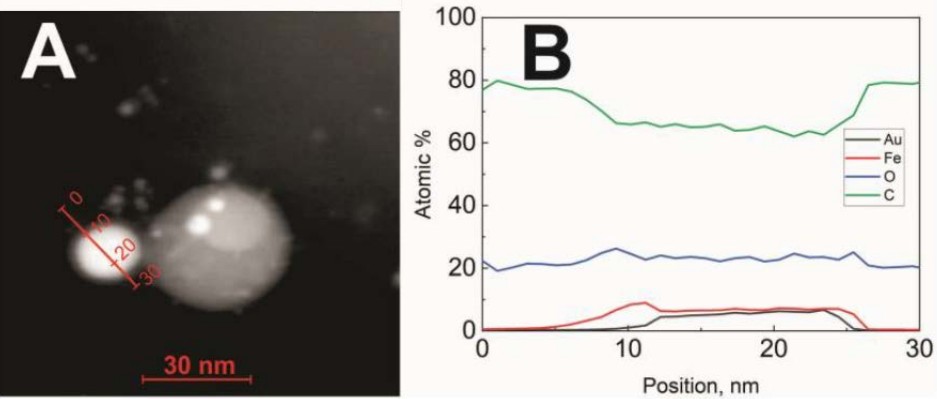

**Figure 4.** TEM image of the analyzed nanoparticle (**A**); EDS analysis of the composition of magneto-plasmonic core-shell nanoparticles generated from Fe50/Au150/Fe25 targets (**B**).

The magnetization of the sample was tested with an external magnet attached to the side of the bottle with the laser-ablated sample. The result is presented in Figure S1. In part B, the majority of nanoparticles are attracted to the side of the bottle. However, some nanoparticles remain in solution. These might be separate gold nanoparticles or very small iron oxide nanoparticles that are not affected by an external magnetic field. The magnetization of the sample was measured as described in our previous work [46]. The experimental results and Brillouin function approximation are presented in Figure S2. It revealed that the sample is weakly magnetic. Obtained parameters: coercivity ~26 mT; saturations of mass magnetization ~2.7 emu/g; remanent magnetization ~1 emu/g. The magnetization is divided by the mass of the sample, which includes the gold part as well, resulting in weak residual magnetization. High coercivity is obtained due to larger (200–300 nm) nanoparticles.

### 3.2. UV-VIS Spectroscopic Analysis

The extinction spectra of the nanoparticle solution obtained from two-layered (Figure 5A) or three-layered (Figure 5B) systems show a distinct plasmon resonance band characteristic for gold nanoparticles, with the maximum at a 509–528 nm interval. The optical extinction spectrum corresponds to the absorption of spherical Au nanoparticles

of size 10–60 nm [47,48]. In addition to this, the general rise of the background going to the shorter wavelengths is visible. It can be attributed to the scattering effect of the nonplasmonic iron nanoparticles. This rise was also clearly observed in our previous work related to the synthesis and analysis of magneto-plasmonic nanoparticles [46]. The plasmonic peaks in a two-layer system are more distinguished from the background in comparison to a three-layered system (Figure 5). This could be due to a larger amount of Fe nanoparticles produced from a three-layer target that creates a higher background and hides part of the plasmonic band. The slight shift of the plasmonic resonance band to a lower wavelength in the case of nanoparticles prepared from three-layered coatings might be related to the smaller diameter of the nanoparticles and/or, to some extent, the formation of AuFe nanoalloys [34].

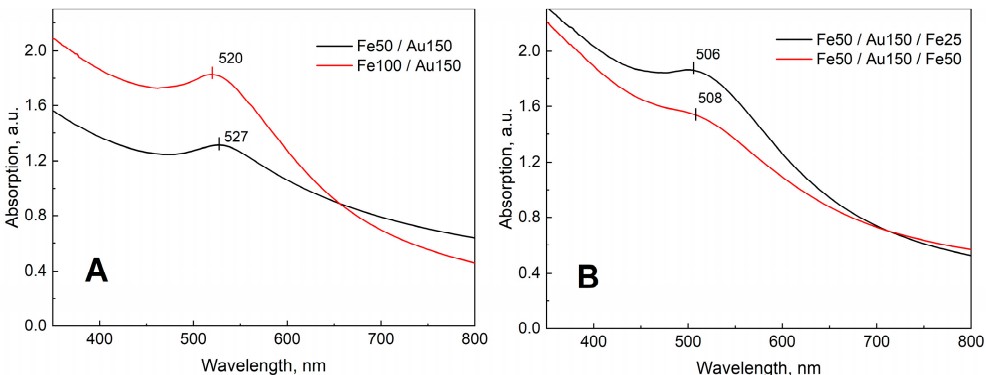

**Figure 5.** UV-VIS extinction spectra of magneto-plasmonic nanoparticles obtained by ablation of two-layer (**A**) and three-layer (**B**) metal films in acetone in the wavelength range 350−800 nm.

### 3.3. SERS of Magneto-Plasmonic Nanoparticles

Prior to the analysis of the adsorbed probe biomolecule adenine, we recorded SERS spectra of bare magneto-plasmonic nanoparticles (Figure 6). The SERS spectra show broad spectral features in the vicinity of 1586–1591 and 1328–1360 $cm^{-1}$ which are characteristic of carbon material G and D vibrational bands, respectively [49–52]. The intensity of these bands is resonantly enhanced; therefore, even a small amount of carbon material may result in relatively intense Raman features. This carbon material was most likely created during the laser-ablation process in acetone. Similar bands were previously observed in the Raman spectra of magneto-plasmonic nanoparticles prepared by laser ablation in organic solvents [53]. We found that the relative intensity of these bands varied from sample to sample. We were not able to connect the relative intensity of carbon bands with the composition of films used for laser ablation. The clearly defined band at 667–674 $cm^{-1}$ signifies the presence of magnetite ($Fe_3O_4$) at the surface of hybrid nanoparticles [54,55]. This band was assigned to the $A_{1g}$ symmetry mode associated with the symmetric stretching vibration of Fe-O bonds [32]. Importantly, no bands characteristic of maghemite ($\gamma$-$Fe_2O_3$, broad peaks at 350, 500, and 700 $cm^{-1}$) or hematite ($\alpha$-$Fe_2O_3$, strong and narrow bands at 412 and 290 $cm^{-1}$) structures are visible in the spectrum, indicating the presence of the pure magnetite phase [56]. One can see that the relative intensity of the magnetite band compared with carbon features is slightly higher in the case of a sample prepared with a higher amount of Fe in the initial coating (a three-layer film) (Figure 6). The discussed spectral features are distinct for all synthesized nanoparticles, irrelevant to the ablation target used. Only subtle variations appear in the parameters of carbon compounds and magnetite bands.

The enhancement factor (*EF*) for these nanoparticles was calculated using the SERS reporter molecule 4-MBA [57,58]. The obtained value for the 632.8 nm excitation wavelength is 5.8 ($\pm$2.8) $\times 10^4$; the calculation procedure is presented in Supporting Information. More than ten times lower *EF* was obtained for the 785 and 830 nm excitation wavelengths (Figure S3). Shumskaya et al. indicated that an *EF* of the order of $10^4$ is sufficient for SERS

applications in the construction of chemo- and biosensors [59]. Such an *EF* was estimated for Ni/Au core-shell magneto-plasmonic nanoparticles by using Methylene Blue dye as a test analyte [59]. Li et al. reported an *EF* of $1.1 \times 10^5$ for $Fe_3O_4$/Au nanostructures by analysis of SERS spectra from malachite green dye [60]. The magneto-plasmonic $Fe_3O_4$/Au composites prepared by the solvent-thermal method exhibited SERS analytical *EF* exceeding $2 \times 10^5$ for the analysis of 4-nitrothiophenol [61]. Ye et al. developed silicon-based substrates with microarrays where magneto-plasmonic $Fe_3O_4$/Au nanoparticles were assembled for SERS analysis of rhodamine 6G dye [62]. Such structures gave SERS EFs higher than $10^6$. Hu et al. reported on the possibility of tuning SERS *EF* from $10^4$ to $10^7$ by using liquid substrates containing suspensions of $Fe_3O_4$ /Au nanoparticles [63]. It was found that magneto-plasmonic $Fe_2O_3$/Au nanoparticles are able to provide SERS *EF* around $10^5$ by using 2-naphthalenethiol as a probe molecule [64].

The structure of magneto-plasmonic nanoparticles obtained in this work is different compared with laser-ablation synthesized nanoparticles in acetone from bulk Fe-Au tar-get [29]. Wagener et al. demonstrated the solvent-controlled phase structure of nanoparticles; an iron-gold core-shell structure was obtained in acetone [29]. The mechanism of the laser ablation process in liquids is very complex and still under extensive development [65,66]. The difference might be associated with the laser treatment of thin coatings in our study and the application of a picosecond-pulsed laser instead of the femtosecond-pulsed radiation employed in our work [65]. TEM images (Figure 4) clearly show lower electronic contrast for the shell of nanoparticles compared with the core, which has a lower electronic density typical for iron oxides [53].

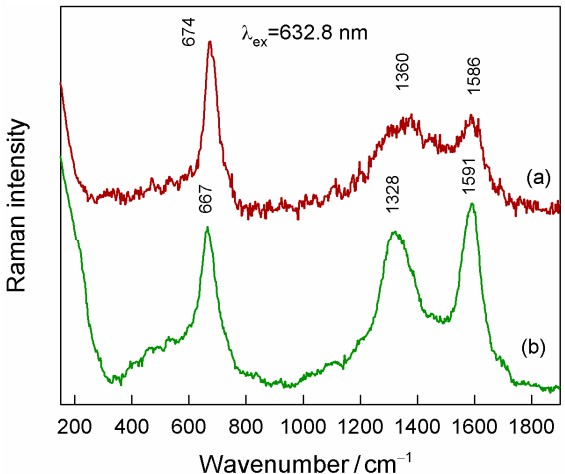

**Figure 6.** SERS spectra of bare magneto-plasmonic nanoparticles prepared by laser-ablation from (a) Fe50/Au150/Fe25 and (b) Fe50/Au150 films. Intensities are normalized to the intensity of the $Fe_3O_4$ band near 674/666 $cm^{-1}$. The excitation wavelength is 632.8 nm (0.8 mW).

*3.4. SERS of Adenine Adsorbed at Magneto-Plasmonic Nanoparticles*

Plasmonic properties of nanoparticles were verified by employing SERS spectroscopy of adsorbed adenine (Ade) as a probe biomolecule [67]. For this, SERS spectra of adenine adsorbed from a 0.1 mM water solution on a magneto-plasmonic nanoparticle substrate were measured. Further detailed SERS studies using adenine solution revealed that the highest SERS signal was obtained using the SERS substrate prepared with laser ablation synthesized from Fe100/Au150 and Fe50/Au150 coatings. This might be due to the composition of the obtained nanoparticles. In both of these samples, the amount of gold was relatively higher than that of iron. Figure 7 shows the SERS spectra of adsorbed Ade on magneto-plasmonic nanoparticles with the SERS spectra of this probe ligand on laser-ablated Au nanoparticles. For comparison, Raman spectra of the solid-state form and aqueous solutions of Ade prepared with $H_2O$ and $D_2O$ solvents as well as an acidic water solution are demonstrated in Figure 8. Two bands marked by the star at 981 and

$1051 \text{ cm}^{-1}$ are associated with stretching vibrations of solution $SO_4^{2-}$ and $HSO_4^{-}$ ions, respectively [68]. The positions of vibrational bands and assignments are listed in Table 2. The clearly resolved band at $683 \text{ cm}^{-1}$ visible in SERS spectra (Figure 7) belongs to $Fe_3O_4$. The presence of this band confirms the preservation of magnetite structure upon adsorption of the probe biomolecule. Adenine possesses multiple adsorption sites (ring $\pi$ system, ring nitrogens, and $NH_2$ group) (Figure 9) and can be positively or negatively charged depending on protonation at the N1 site ($pK_1 = 4.1$) or ring deprotonation ($pKa = 9.8$) [69,70]. The most intense band in the SERS spectra at $736 \text{ cm}^{-1}$ corresponds to adenine ring breathing vibration [71–76]. The corresponding band in the Raman spectrum of adenine powder appears at a considerably lower frequency ($723 \text{ cm}^{-1}$). Such a shift is characteristic of adsorbed adenine on the Au surface [69,71]. This mode downshifts to $707 \text{ cm}^{-1}$ upon labile hydrogens' exchange to deuterons in $D_2O$ solution (Figure 8). Comparison of Ade solution spectra at pH 6.2 and 1.5 reveals a shift of the $1486 \text{ cm}^{-1}$ band to $1408 \text{ cm}^{-1}$, the $1332 \text{ cm}^{-1}$ band to $1312 \text{ cm}^{-1}$, the absence of the $1251 \text{ cm}^{-1}$ band, and a shift of the $623 \text{ cm}^{-1}$ band to $617 \text{ cm}^{-1}$ (Figure 8). In SERS spectra, the bands at $1454–1464 \text{ cm}^{-1}$ (which corresponds to the Ade solution band at $1486 \text{ cm}^{-1}$), $1342–1348 \text{ cm}^{-1}$ (the solution band at $1332 \text{ cm}^{-1}$), $1236–1240 \text{ cm}^{-1}$ (the solution band at $1251 \text{ cm}^{-1}$), and $627–629 \text{ cm}^{-1}$ (the solution band at $622 \text{ cm}^{-1}$) are visible, indicating a neutral form of the adsorbed Ade ring. DFT modeling suggested an increase in the frequency of ring breathing mode from 725 to $743 \text{ cm}^{-1}$ upon bonding of the N3 site with the $Au^+$ ion, while an upshift of only $2 \text{ cm}^{-1}$ was demonstrated upon metal binding to the N7 site [69]. Thus, our experimental SERS data are consistent with the strong interaction of the Ade ring with the surface through the N3 atom. However, the involvement of the N9 atom with surface bonding is also possible, as was suggested previously [71].

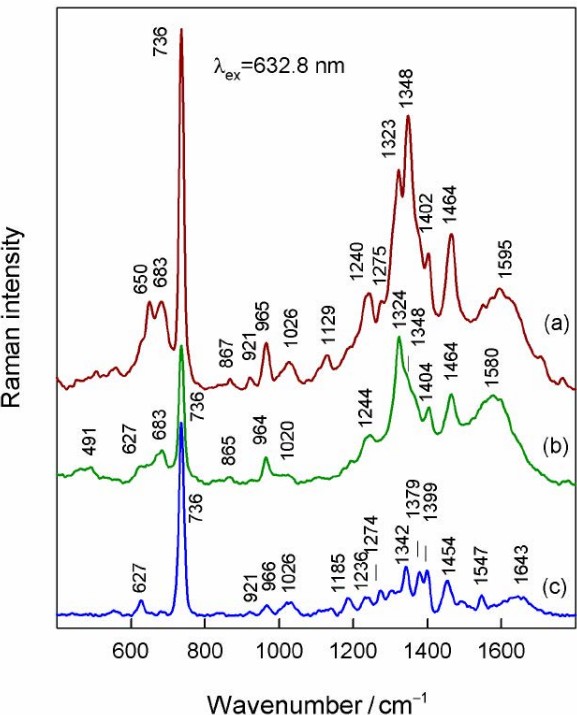

**Figure 7.** SERS spectra of adenine adsorbed from 0.1 mM aqueous solution on magneto-plasmonic nanoparticle substrate and on gold nanoparticle substrate produced by laser ablation of (a) Fe100/Au150, (b) Fe50/Au150, and (c) bulk Au target. The excitation wavelength is 632.8 nm (2.5 mW).

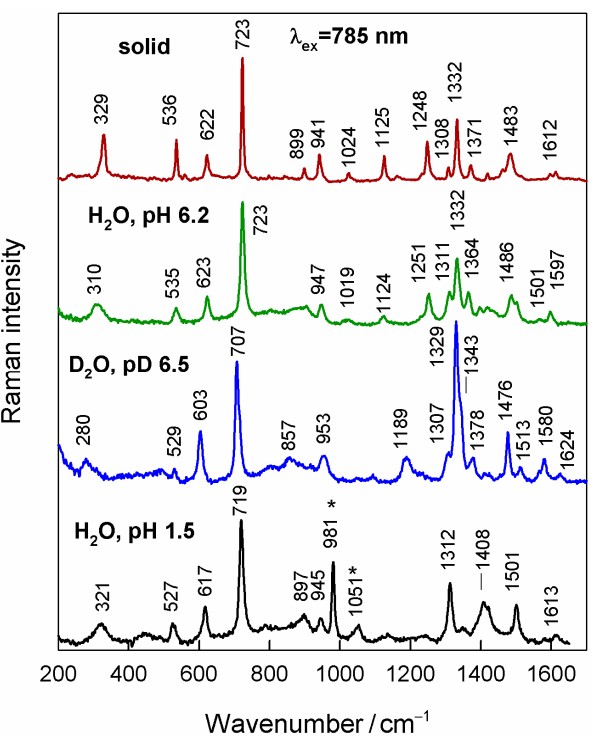

**Figure 8.** Raman spectra of solid adenine and solutions (7 mM) prepared with $H_2O$ (pH 6.2), $D_2O$ (pD 6.5), and acidic (pH 1.5) $H_2O$ solution. The excitation wavelength is 785 nm. The stars denote the bands that originated from $SO_4^{2-}$ (981 $cm^{-1}$) and $HSO_4^{-}$ (1051 $cm^{-1}$) species.

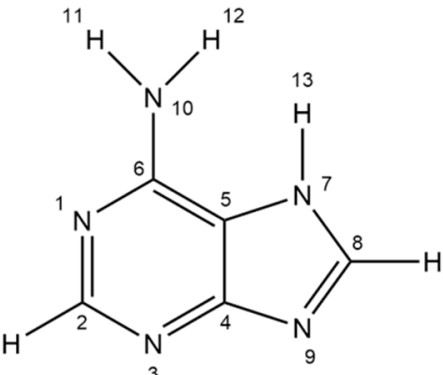

**Figure 9.** Molecular structure and atom labeling of adenine tautomer N7H.

A comparison of SERS spectra from adsorbed Ade on laser-ablated Au and magneto-plasmonic nanoparticles reveals considerable differences in the relative intensities and frequencies of prominent bands (Figure 7). The higher relative intensity of ring breathing mode near 736 $cm^{-1}$ is consistent with a more perpendicular orientation of the ring plane with respect to the surface for Ade adsorbed on laser-ablated Au nanoparticles. The band at 964–966 $cm^{-1}$ was assigned to the rocking $NH_2$ vibrational mode coupled with stretching of the N1-C6 bond and deformation of the N7-C8-N9 group (r($NH_2$) + ν(N1-C6) + δ(N7-C8-N9)) (Table 2). An increase in the relative intensity of this band indicates an increase in the angle between the Ade ring plane and the surface normal. The clearly defined SERS band 1454–1464 $cm^{-1}$ was assigned to the ν(N1-C6) + β(C2H) + ν(C2-N3) + δ($NH_2$) vibrational mode (Table 2). In the Ade solid-state spectrum, this band appears at 1483 $cm^{-1}$ and shifts to 1476 $cm^{-1}$ in $D_2O$ solutions (Figure 8). The frequency of this band differs considerably when comparing SERS spectra on Au (1454 $cm^{-1}$) and magneto-plasmonic (1464 $cm^{-1}$) nanoparticles. Such a frequency shift suggests the involvement of the $NH_2$ group in the interaction of Ade with a surface of magneto-plasmonic nanoparticles. Similar

intensification of high-frequency bands in the vicinity of 1300–1400 cm$^{-1}$ compared with ring breathing mode at 735 cm$^{-1}$ was observed in the SERS spectra of Ade adsorbed on Ni and Ni-Ag nanoparticles [77]. Based on density functional theory (DFT) analysis and the similarity of spectra observed at Ni and Ni-Ag surfaces, it was suggested that Ade primarily interacts with Ni adsorption sites [77].

**Table 2.** Experimental and calculated vibrational frequencies of adenine and model surface complexes and assignments of the bands.

| Solid State | Solution $H_2O$ ($D_2O$) | SERS Au | SERS Au-$Fe_3O_4$ | Calc. Ade (N7H) | Calc. $Au_3$-Ade(N7H) | Calc. $Fe_3O_4$-Ade(N7H) A | Calc. $Fe_3O_4$-Ade(N7H) B | Assignments |
|---|---|---|---|---|---|---|---|---|
| 1483 m | 1486 m (1476 m) | 1454 m | 1464 s | 1509 | 1488 | 1495 | 1503 | $\beta$(C2H), $\nu$(N1-C6), $\nu$(C2-N3), $\delta$(NH$_2$) |
| 1371 w | 1364 m (1378 w) | 1399 w | 1402 w | 1405 | 1383 | 1396 | 1390 | $\nu$(C4-C5), $\beta$(CH$_2$) |
| 1332 s | 1332 s (1329 vs) | 1342 m | 1348 | 1365 | 1337 | 1342 | 1334 | $\nu$(C2-N3), $\nu$(C8-N9), $\beta$(C2H) |
| 1248 m | 1251 m (1189 m) | 1236 | 1240 | 1226 | 1226 | 1228 | 1240 | $\nu$(C2-N3), $\beta$(C8H), $\nu$(C8-N9) |
| 941 m | 947 m (953 m) | 966 w | 965 m | 948 | 979 | 985 | 981 | $\delta$(N7-C8-N9), $\nu$(N1-C6), r(NH$_2$) |
| 723 vs | 723 vs (707 vs) | 736 vs | 736 vs | 726 | 732 | 725, 736 | 715 | ring breathing, $\nu$(Fe-O) |
| 622 m | 623 m (603 m) | 627 sh | 629 sh | 618 | 621 | 627 | 686 | $\nu$(C5-C6), $\beta$(R1), $\beta$(R2) |
| | | | | | | | 371 | $\nu$(Fe-N10) |
| | | | 295 m, br | | | 243, 263 | | $\nu$(Fe-N3), $\nu$(Fe-N9), $\nu$(Fe-O) |
| | | | 218 w, br | | | 151, 208, 216 | | $\nu$(Fe-N3), $\nu$(Fe-N9) |
| | | 200 m, br | | | 187 | | | $\nu$(Au-N3), $\nu$(Au-N9) |

Abbreviations: $\nu$, stretching; $\delta$, deformation; $\beta$, in-plane bending; R1, six-membered ring; R2, five-membered ring; vs, very strong; m, middle, w, weak; br, broad; sh, shoulder.

Analysis of low-frequency SERS spectra reveals insights into the bonding of Ade to laser-ablated Au and magneto-plasmonic nanoparticles (Figure 10). Spectrum from bare magneto-plasmonic nanoparticles before adsorption of Ade does not show any clear low-frequency vibrational mode. However, the broad low-frequency band at 200 cm$^{-1}$ is visible in the spectrum of Ade adsorbed on laser-ablated Au nanoparticles (Figure 10a). A different SERS spectrum was observed for Ade adsorbed on a substrate prepared from magneto-plasmonic nanoparticles; a broad band centered at 295 cm$^{-1}$ became clearly visible along with another lower intensity feature near 218 cm$^{-1}$. We suggest these low-frequency modes are related to the bonding of Ade nitrogens with $Fe_3O_4$ structures.

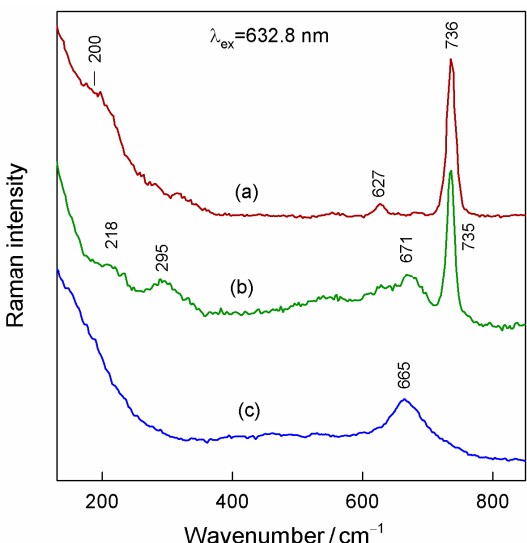

**Figure 10.** SERS spectra in the low-frequency region (130–850 cm$^{-1}$). (a) adenine adsorbed at Au NPs prepared by laser ablation, (b) adenine adsorbed at Fe-Au nanoparticles prepared by laser ablation of Fe50/Au150/Fe25 film, and (c) SERS spectrum of bare Fe-Au nanoparticles prepared by laser ablation of Fe50/Au150/Fe25 film. The excitation wavelength is 632.8 nm.

*3.5. DFT Modelling of SERS Spectra*

To gain more insights into the interaction of adenine with magneto-plasmonic nanoparticles, DFT modeling of the Raman spectra of adsorption complexes was conducted. Figure 11 shows optimized structures of Ade (tautomer N7H) and adsorption complexes of Ade with the Au$_3$ cluster and Fe$_3$O$_4$. Two adsorption sites, through N3/N9 atoms and N10/N7H, were modeled for interaction with the magnetite surface. The calculated Raman spectra are shown in Figure 12. In the case of the Au$_3$-adenine complex, the prominent ring breathing mode shifts to higher wavenumbers, from 726 cm$^{-1}$ (free Ade) to 732 cm$^{-1}$. Such a frequency upshift agrees very well with the experimental SERS spectrum (Figure 7). A similar increase in frequency of the ring breathing mode was predicted by the Fe$_3$O$_4$-adenine (A) complex. In this case, two bands associated with ring breathing mode coupled with Fe-O stretching are visible at 725/736 cm$^{-1}$. However, ring breathing mode was found at considerably lower wavenumbers (715 cm$^{-1}$) in the case of the Fe$_3$O$_4$-adenine (B) complex. Thus, the bonding of Ade with N10/N7H sites does not predict the experimentally observed shift of this mode, suggesting that the major interaction site of Ade with Fe$_3$O$_4$ corresponds to the bonding of Fe with N3/N9 atoms. DFT calculations predict a considerably stronger interaction of Fe$_3$O$_4$ with the N3 site compared with N9 because of the noticeably shorter Fe-N3 bond length (201.3 pm) compared with Fe-N9 (208.1 pm) (Figure 11).

The DFT calculations predict the downshift of the high-frequency band at 1509 cm$^{-1}$ (free Ade) to 1488 and 1495 cm$^{-1}$ upon bonding adenine with the Au$_3$ cluster or Fe$_3$O$_4$ through the N3/N9 sites, respectively (Table 2), while only a small shift (6 cm$^{-1}$) is predicted for the Fe$_3$O$_4$-Ade (B) complex. In experimental spectra, a significant decrease (19–29 cm$^{-1}$) in frequency was detected (Table 2). The band at 941 cm$^{-1}$ (solid-state Ade spectrum) was found to be sensitive to the interaction between the Ade rings. The frequency of this band increased by 6 cm$^{-1}$ in the solution spectrum (Figure 8, Table 2). A high upshift (24–25 cm$^{-1}$) frequency of this band was detected in SERS spectra (Table 2). In agreement with experimental data, DFT modeling predicts an upshift frequency of this mode of 31–37 cm$^{-1}$ upon the formation of surface complexes.

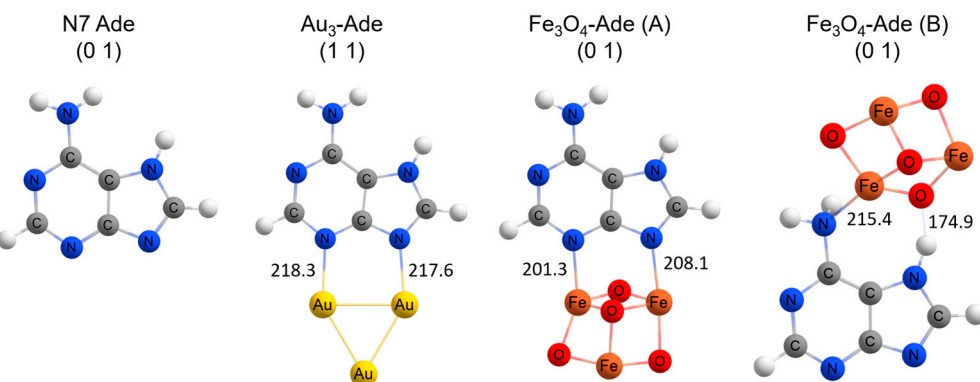

**Figure 11.** Optimized structures of N7H adenine tautomer, Au$_3$-adenine complex, and Fe$_3$O$_4$-adenine complexes formed at N3-/N9- and N10-/N7H- interaction sites, Fe$_3$O$_4$-Ade (A) and Fe$_3$O$_4$-Ade (B), respectively. Numbers in brackets indicate charge and multiplicity. The lengths of chemical bonds are indicated in pm.

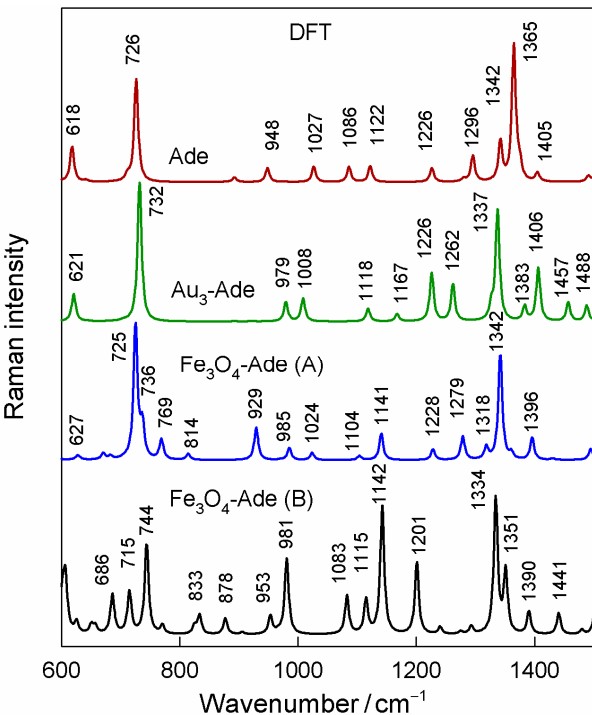

**Figure 12.** Calculated Raman spectra of N7H adenine tautomer, Au$_3$-adenine complex, and Fe$_3$O$_4$-adenine complexes formed at N3-/N9- and N10-/N7H- interaction sites, Fe$_3$O$_4$-Ade (A) and Fe$_3$O$_4$-Ade (B), respectively.

DFT analysis of low-frequency vibrational modes predicts the position of the metal-adsorbate Au-N3 vibrational mode at 187 cm$^{-1}$ (Table 2). This confirms the origin of the experimentally observed broad SERS band near 200 cm$^{-1}$ from adenine adsorbed on laser-ablated Au nanoparticles as associated with Au-N stretching vibration (Figure 10). Peak position coincides well with previously reported DFT analysis of stretching vibration of the Au-N3 bond (196 cm$^{-1}$) for the adsorption complex Ade-Au$^+$ [69]. In the case of hybrid magneto-plasmonic nanoparticles, the broad low-frequency bands were observed at different wavenumbers, i.e., 218 and 295 cm$^{-1}$. DFT modeling suggests that these bands originate from Fe-N stretching vibrations associated with N3 and N9 atoms (Table 2).

## 4. Conclusions

In this work, we have obtained chemically clean magneto-plasmonic nanoparticles from layered Fe/Au and Fe/Au/Fe thin film coatings by applying 1064 nm picosecond laser

ablation in acetone. The magnetic properties of nanoparticles were used to extract them from acetone and arrange them on an aluminum substrate for SERS applications. Based on TEM, EDS, and Raman spectroscopy data, we demonstrated that hybrid nanoparticles consisted of a plasmonic (Au) core and a magnetic ($Fe_3O_4$) shell. Hybrid magneto-plasmonic nanoparticles exhibited distinct plasmon resonance bands characteristic of spherical gold colloids, with a maximum at 509–528 nm. SERS spectra revealed that the probe molecule-adenine, adsorbs on the magnetite site instead of gold, suggesting that the magnetic shell is sufficiently compact to prevent penetration of adenine into the gold core. The interaction of adenine with magnetite was also confirmed by DFT calculations. A strong SERS signal from adsorbed adenine demonstrated that the magnetic shell does not markedly diminish the electromagnetic enhancement provided by the plasmonic core. A magnetic shell can serve for manipulation and arrangement of nanoparticles at a chosen surface, while a plasmonic core can ensure vibrational spectroscopy sensing of nucleic acid bases and other biomolecules. Because the magnetic shell (magnetite, $Fe_3O_4$) is biocompatible, hybrid nanoparticles can be employed in biomedicine applications [78].

**Supplementary Materials:** The following supporting information can be downloaded at https://www.mdpi.com/article/10.3390/coatings13091523/s1. Figure S1: Solution of ablated magneto-plasmonic nanoparticles in acetone (A) just prepared and (B) in 15 min after exposure to a permanent magnet; Figure S2: Hysteresis loop of the laser ablated magneto-plasmonic nanoparticles (Fe50/Au150/Fe25) (black) and approximation with the Brillouin function (red); description of enhancement factor calculations; Figure S3: (A) SERS spectra of a 4-MBA molecule obtained using magneto-plasmonic nanoparticles at 633, 785, and 830 nm laser radiations. Shaded areas represent the standard deviation from nine measurements. (B) SERS enhancement factors calculated for different laser excitations.

**Author Contributions:** L.M.: investigation, methodology, writing–original preparation, writing–review, and editing; E.S.: methodology, conceptualization, formal analysis; S.A.-G.: investigation, methodology, visualization; V.P.: investigation, visualization, formal analysis; R.T.: visualization, formal analysis, methodology; M.T.: investigation, methodology, formal analysis, DFT calculation; M.S.: investigation, methodology, formal analysis; A.Z.: methodology, formal analysis, visualization; A.S.: investigation, methodology, formal analysis; V.Š.: methodology, conceptualization, supervision; G.N.: conceptualization, supervision, writing—original draft preparation, writing—review, and editing. All authors have read and agreed to the published version of the manuscript.

**Funding:** This work has received funding from the European Regional Development Fund (Project No. 01.2.2-LMT-K-718-03-0078) under a grant agreement with the Research Council of Lithuania (LMTLT).

**Institutional Review Board Statement:** Not applicable.

**Informed Consent Statement:** Not applicable.

**Data Availability Statement:** Not applicable.

**Acknowledgments:** The authors gratefully acknowledge the Center of Spectroscopic Characterization of Materials and Electronic/Molecular Processes (SPECTROVERSUM Infrastructure) for the use of Raman spectrometers. Authors thank Voitech Stankevic and PhD student Jorunas Dobilas for magnetization measurements.

**Conflicts of Interest:** The authors declare no conflict of interest.

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
