# Peer review of "Magneto-Plasmonic Nanoparticles Generated by Laser Ablation of Layered Fe/Au and Fe/Au/Fe Composite Films for SERS Application"

_coatings, doi:10.3390/coatings13091523_

Round 1

Reviewer 1 Report

The paper reported magneto-plasmonic nanoparticles fabricated by using picosecond-pulsed laser for ablation of Fe/Au and Fe/Au/Fe composite thin films in acetone. Hybrid nanoparticles were arranged on an aluminum substrate by magnetic field for application in SERS. The hybrid nanoparticles can be applied in the construction of biosensors and biomedicine applications. Some of the listed points should be revised for further improving the manuscript and publication.

1. The references for research status are suggested to be updated to the last three years.

2. I suggest that the author include a discussion on relevant papers including Colloids and Surfaces A: Physicochemical and Engineering Aspects, 2023, 671, 131661; Biosensors, 2023, 13(5), 530; Optics & Laser Technology, 2023, 163, 109429; SCIENCE CHINA Technological Sciences, 2023, 66, 853-862; Materials, 2023, 16(8), 3083.

3. Magneto-plasmonic nanoparticles generated from Fe50/Au150 target was core-shell nanoparticles with the core of Au and the shell of Fe. What was the nanoparticles generated from Fe/Au/Fe target.

4. Please calculate the enhancement factor (EF) of SERS of adenine adsorbed at magneto-plasmonic nanoparticles, compare the EF with that previously reported, and explain the advantages of this work for SERS application.

5. According to Figure 7, the Raman enhancement effect of adenine adsorbed at Fe50/Au150 was weak than that at bulk Au target, but the Raman enhancement effect of adenine adsorbed at Fe100/Au150 was strongest. Please explain the role of Fe in SERS and why did different Fe content have different SERS result.

Reviewer 2 Report

In this manuscript, the authors proposed Fe/Au nanoparticles prepared by laser ablation for SERS and biomedical applications. Three different layer structured targets were employed to fabricate Fe/Au composites, and the SERS activities of prepared samples were analyzed in the paper. While the paper is quite clear, it could be further improved by the following suggestions and discussions.

1) Authors briefly introduced the preparation of nanoparticles, and I would like to know more details about that. The sample target with different layer structures were prepared by sputtering. I wonder if the Fe layer deposited after sputtering is pure Fe or iron oxide and how the nanoparticles are treated after ablation, especially if applied heat procedure. Because the resulted samples are pure Fe3O4 shell, which is different from the conclusion of other study. In paper “Solvent-surface interactions control the phase structure in laser-generated iron-gold core-shell nanoparticles”, their group found Fe3O4 shell was only formed in wafer base and Fe was formed in acetone.

2) The SEM and TEM analysis in the manuscript showed the nanoparticles from only one type of target. I am interested to see the TME images of nanoparticles ablated from other structures like Fe/Au/Fe.

Also in the later discussion, authors mentioned three-layer target formed larger amount of Fe. I wonder the concentration difference of Fe in three- and two-layer target structures. Because from target configuration, Fe50/Au150 does not have huge Fe concentration difference compared to Fe50/Au150/Fe25. Providing the concentration of various type of samples would be better to help understanding.

Similar issue needs to be clarified in the 3.3. SERS of magneto-plasmonic nanoparticles part, carbon spectra is more distinguishable in the three-layer structure from Figure 6.

3) In the SERS of adenine analysis part, authors represented a comparison of spectra of nanoparticle substrate and pure Au substrate in Figure 7. Regarding of the proposed application and title of this paper is SERS, I would suggest authors provide more details about SERS application. For example, advantages as enhancement of SERS signal based on the nanoparticle substrate, which could be concluded from Figure 7. And if more experiments about SERS testing could be provided like sensitivity, recyclable performance, that would be also helpful to indicate the SERS application.

The last part about interaction between adenine and prepared nanoparticles shows the potential utilization as biosensor or biomedical therapy, however, it would be better to state and emphasize the connection between this part and applications.  

Reviewer 3 Report

Authors present magnetoplasmonic nanoparticles made via laser ablation. Nanoparticles can be used as a SERS platform, and due to Au-Fe structure, they can be arranged on the surface of aluminium foil with NdFeB magnet. The nanoparticles are free from organic stabilizers and because of biocompatibility of magnetic shell and SERS activity by plasmonic gold core can be used as a biosensors. The manuscript can be accepted for publication, however some issues should be explained:

- the literature survey is good, however it should be mentioned that SERS platforms are also manufactured without liquid layer (in the air) via e.g. femtosecond lasers:

https://www.sciencedirect.com/science/article/pii/S2238785421003100

- in Figure 2 the scale should be white or black, the red color is hard to read.

- please explain why acetone was chosen for the experiment. Did Authors tested other solvents or water?

- to compare the experiment with others the fluence should be given.

- could Authors calculate Enhancement Factor (EF) for this system? It allow to compare SERS platform with others (what should be done in Conclusions). For calculating EF please do experiment with the use of pMBA. 

The language is fine, I could not find any big mistakes. Some error checking would be good before re-submitting.

Reviewer 4 Report

Production of the magneto-plasmonic nanoparticles on the multigram scale is of great demand considering different biomedical applications. Laser technologies especially laser ablation in liquids could fit all the requirement with high productivity and chemical clearness of the resulted nanoparticles. From this poin of view the manuscript under consideration is on a cutting edge of the laser science R&D. However, there is an exponential growth of the researcher and project dealing with LAL including Au-Fe bimetal nanoparticles. The only new concept introduced by the authors is picosecond regime used to generate nanoparticles, that is rather unpractical since nanosecond LAL regime yields in higher productivity with cheaper setup. To retain a publication interest authors should provide more explanation for the observed results, which are quite interest if it is not just an experimental mistake or so on. 

For instance, here some questionable concepts which can be find through the manuscript:

1) Why LAL in acetone resulted in the formation of Fe3O4 rather than a simple AuFe nanoalloy? In accordance with well known nanoscale phase diagram for the LAL of Au-Fe coatings and layered films, Fe3O4 production takes place in the case of ablation in water rather than acetone.

2) Target preparation process could be a possible source of the oxygen which results in Fe3O4 formation. It is known that Fe layers are very sensitive to air exposure. From this point of view authors should try Au/Fe/Au or Fe/Au (with upper Au layer) multilayer target to minimize Fe oxidation and enhance the formation of Au-Fe core-shell nanoparticles rather than Au-Fe3O4.

3) Why authors claimed formation of Au (core) - Fe (shell) structure? In Accordance with Raman investigation the resulted nanoparticels contain large fraction of Fe3O4. In addition, EDS analysis clearly demonstrate that Fe atoms are present in both shell and core.

4) It is not clear what is the magnetic state of the resulted nanoparticles. Fe3O4 and Fe obviously have different magnetic moments and thus different orientation on the substrates during SERS measurement. In fact, Fe3O4-Au nanoparticles are not so interested for biomedical application. 

5) To present a correct experiment authors must compare pure Fe nanoparticles in addition to Au-Fe and Au ones. Moreover, if the only resulted products are Fe3O4-Au nanoparticles, why not to use initial targets with Fe3O4 and Au layers?

6) I did not find any nanoparticles size distibution including dependencies on both targets composition and thickness. XRD data could be very helpful in dealing with a dilema of Fe versus Fe3O4 shells.

7) Why authors call resulted nanoparticles as a hybrid while they are bimetallic (Au-Fe). The term "hybrid" usually denotes, for example metal-semiconductor nanoparticles that is true for Fe3O4-Au ones.

8) Please add C and O2 distribution in the EDS analysis as well. It is not clear at all where carbon related G and D bands and Fe3O4 formation came from. 

9) There is no labeled plasmon for the bottom curve in the Figure 5(b).

10) Sers spectra in the Figure 6 should start below 200 cm-1 as in the Figure 10 for clarity in determing the nature of the observed Raman bands.

11) What about the real numers behind SERS? How exactly high is enhancement? What about a chemical activity of the Fe and Au during SERS expweriment and how can you estimate chemical enhancing of the SERS signal in this case?

12) The phrase "SERS spectra revealed that probe molecule-adenine adsorbs on magnetite site instead of gold. This was also confirmed by DFT calculations." is meaningless, since LAL resulted in the production of AuFe nanoparticles with Fe3O4 shells alone and there are no Au sites at all for adsorbing on. 

13) The bottom spectrum in Figure 6 clearly demonstrates T2g band of Fe3O4 in the 400-600 cm-1 range. Why did it appear in the case of target with lower Fe concentration?

Round 2

Reviewer 2 Report

The authors have addressed all my concerns and those of the other reviewers very thoroughly and to my satisfaction, and I recommend publication in Coatings.

Author Response

Authors' response:

We thank the Reviewer for the analysis of manuscript and are very grateful for his/her time spent for reviewing the work. We appreciate the positive feedback from the Reviewer.

Reviewer 3 Report

The improvements made by the Authors are fine, the manuscript can be accepted for publication.

Author Response

Author response:

We thank the Reviewer for the analysis of manuscript and are very grateful for his/her time spent for reviewing the work. We appreciate the positive feedback from the Reviewer.

Reviewer 4 Report

Despite authors have adressed most of the questions and suggestions arised, there are still no evidence of magnetic properties of the obtained hybrid nanoparticles, at least in a similar way presented in https://doi.org/10.3390/nano12162860 with magnetic loops and so on. 

In addition, the enhancement factor provided by authors in the revised manuscript shoul be explained.  Namely, at what wavelength do you obtain an EF of 5.8*10^4? What is EF vs wavelength dependence? Please compare obtained EF with priviously published for Fe3O4-Au nanoparticles.

Author Response

Sincerely yours,

Dr. Gediminas Niaura (corresponding author)
